# Leading Role of E-Learning and Blockchain towards Privacy and Security Management: A Study of Electronics Manufacturing Firms

Abdelmohsen A. Nassani [1], Adriana Grigorescu [2,3,*], Zahid Yousaf [4,*], Raluca Andreea Trandafir [5], Asad Javed [6] and Mohamed Haffar [7]

1 Department of Management, College of Business Administration, King Saud University, P.O. Box 71115, Riyadh 11587, Saudi Arabia; nassani@ksu.edu.sa
2 Department of Public Management, Faculty of Public Administration, National University of Political Studies and Public Administration, Expozitiei Boulevard 30A, 012104 Bucharest, Romania
3 Academy of Romanian Scientists, Ilfov Street 3, 050094 Bucharest, Romania
4 Higher Education Department, Government College of Management Sciences, Mansehra 21300, Pakistan
5 Department of Public Administration, Faculty of Law and Public Administration, Ovidius University of Constanța, Mamaia Boulevard 124, 900527 Constanta, Romania; raluca.trandafir@univ-ovidius.ro
6 Department of Management Sciences, Hazara University, Mansehra 21100, Pakistan; asadjaved@hu.edu.pk
7 Department of Management, Birmingham Business School, University of Birmingham, Birmingham B25 2TT, UK; m.haffar@bham.ac.uk
* Correspondence: adriana.grigorescu@snspa.ro (A.G.); muhammadzahid.yusuf@gmail.com (Z.Y.); Tel.:+40-724-253-666 (A.G.); +92-321-980-4474 (Z.Y.)

**Abstract:** The success of businesses is now mostly dependent on e-learning methods as these methods are a rapidly growing innovative technology. Blockchain technology has also been considered to have the ability to change businesses. Therefore, this research aims to explore the direct influence of e-learning on the effectiveness of privacy and security in electronics manufacturing. This study also examines the considerable mediating role of the adoption of blockchain technology between e-learning and privacy and security. Furthermore, the current research investigates how digital orientation moderates the association between e-learning and privacy and security. For the collection of data, the cross-sectional research design and random sampling technique were used, and data were gathered from employees of electronics manufacturing firms in Pakistan through questionnaires. The working response rate of the study was 70%. The findings proved that e-learning plays a considerable role in boosting the privacy and security of electronics manufacturers. The results also demonstrate that the adoption of blockchain technology mediates and digital orientation moderates the link between e-learning and privacy and security. This study adds to the better understanding of management by presenting the significant role of e-learning and blockchain technology in improving the efficiency of privacy and security for electronics manufacturing firms.

**Keywords:** e-learning; blockchain technology adoption; digital orientation; privacy and security; electronics manufacturing firms

## 1. Introduction

Technological developments have made considerable contributions to the advancement of the global economy and the improvement in humanity, for example, through e-learning technology [1]. Because this is helpful in changing how trading deals are performed, e-learning is one of the technical advancements that have a great deal of promise for economic growth [2]. Firm information privacy and security strategies are already being implemented global, and have also become a focal point of research [3]. E-learning is the basic driver of privacy and security of a firm. Rapid economic development that focuses on privacy and security management of the firm has resulted in a considerable rise in

technological issues and concerns among all firms [4]. Accordingly, security management has grown to be a significant factor in leveraging economically encouraging opportunities, and e-learning is considered a primary antecedent of industry advancement, taking a range of technological deliberations into account [5]. E-learning has transformed the means via which consumers use and acquire data, and has replaced conventional systems [6]. Firms are currently increasing their contact, particularly via digital means, using a widespread approach, and adopting blockchain technologies to increase their reach to their neighboring partners [7]. Electronic manufacturing firms are aiming to actively involve themselves in e-learning practices to restructure themselves in terms of the fundamental ideals of blockchain technology [8].

Recognizing that blockchain technology adoption is distinctively taking place universally, it is interesting to monitor how e-learning may interact with blockchain technology, and one of the fascinating things is its ability to boost privacy and security management of electronics manufacturing firms [9]. Using blockchain technology can facilitate the re-thinking of how we increase the privacy and security of operations and carry out business competitively, which is a primary goal of the firm [10], especially electronics manufacturing firms. Blockchain technology refers to the potential of the distributed ledger set-up to deliver considerable advantages, for instance, decentralization, reliability in the distributed structures, secure information storage systems, and zero-exchange transaction costs [11]. It is basically a network of distributed ledgers in which nodes correspond with each other to directly operate transactions and data [12]. Accordingly, electronics manufacturing firms that have higher e-learning capability are significantly reliant on their proficiency to support inventiveness and become engaged in procedures that provide guidance for blockchain technology adoption [13]. A firm's e-learning ability supports the adaptation of changes, and their information about advanced technology directs them to become involved in exercises that lead to the adaptation of blockchain technologies [14]. To address such barriers, this research emphasizes the e-learning capability of electronics manufacturing firms in acquiring knowledge about the latest blockchain technology adoption, which also successfully ensures the security and privacy management of practices in the firm.

Digital orientation relies on e-learning technologies to encourage the latest market strategies which draw the attention of potential customers to security management [15,16]. The rapid application of technologies, plus changing market trends and methods, highlight the significance of e-learning and digitalization [17,18]. Electronics manufacturing firms engaged in successful privacy and security management of different projects emphasize not only e-learning capabilities but also digital orientation. Blockchain technology has gained significant researcher attention since it was first explored as a critical determining factor for the flourishing security management of operations with positive results in electronics manufacturing firms [19]. Digital orientation is a purposeful capability that emphasizes market transformations adapted by electronics manufacturing firms, such as digitalized machinery, tools, robotics, and measures to attain the latest information technology benefits [20]. It is helpful in developing significant activities that assist in security management through the adoption of blockchain technologies [21]. Electronics manufacturing firm management is mostly reliant on digital orientation skills with a vision to address changing trends and discover methods that meet the fundamentals of excellence [22,23].

Most significantly, the focus on e-learning strategic programs has confirmed that digital orientation can assist electronics manufacturing firms in their security management concerns. Studies conducted thus far do not provide any substantial clarification of the considerable role of e-learning, or regarding the adoption of blockchain technology, in response to managing privacy and security of a firm through the significant supportive role of digital orientation, which comprises a noteworthy research gap. Prior research highlights numerous outcomes of e-learning in different sectors, such as the effect of the COVID-19 epidemic on the growth in e-learning [24], the e-learning ecosystem [25], and e-learning cultural challenges [26]. However, the current study will provide a platform for

electronics manufacturing firms to evaluate the effectiveness of their e-learning capability with a viewpoint to successfully achieving privacy and security in the firm.

The empirical model in the research presents three hypotheses. First, H1 suggests that in electronics manufacturing firms, e-learning is positively linked with privacy and security management of information. Second, due to its exceptional growth, the worldwide market has been restructured, and therefore, we investigate how adoption of blockchain technology is supported through the antecedent of e-learning capability, and similarly, how blockchain technology performs a significant mediating role in e-learning and security management in firm practices. Finally, we examine the digital orientation moderation in the association between e-learning and privacy and security of electronics manufacturing firms. Thus, the current research provides an interesting model that presents critical implications for policymakers and management, and suggests future directions for further studies that enlighten the consequent rapid changes. The paper format is as follows. The next section presents a literature review and hypotheses formulation. In the following section, methodology and data collection procedures are presented. The Section 4 comprises analysis of the results, and the last section presents a discussion of the study.

## 2. Literature Review

Operational definitions of all the variables of interest are given below:

### 2.1. E-Learning

E-learning refers to the formalized delivery of learning and training via digital resources such as tablets, cellular phones, and computers that are connected to the Internet. Although e-learning refers to the acquisition of knowledge that takes place via electronic technologies [27], it has several advantages, including enhancing supply chain transparency, improving data security, gaining efficiency by eliminating intermediaries, and reducing costs. However, it also has some drawbacks, including dealing with complex technology, difficulty in integrating old systems with blockchain, and a lack of regulations.

### 2.2. Adoption of Blockchain-Technology

Blockchain technology is a complex system of shared and immutable ledgers, meaning that it is difficult to hack, cheat, and change recorded information [28]. It is basically a digital ledger that helps in recording transactions and is distributed across a network of computer systems. Due to this primary feature, blockchain has the potential to be used across numerous industries [29]. Public administration is sensitive to data security, and the adoption of blockchain is an important stage of digitalization [30]. However, this technology was initially adopted by highly digitalized companies (such as e-commerce) and has now expanded to other industries, including agriculture [31].

### 2.3. Privacy and Security

Privacy refers to the exact control over how an individual's data and information are used and viewed. It concerns an entity's ability to decide for themselves when, for what reason, and how their personal information is handled by outsiders. Security refers to protection against danger and threats, and the provision of human safety, self-determination, and human dignity [32].

### 2.4. Digital Orientation

Digital orientation indicates a business that focuses on the digital market, covering strategies related to the use of digital technologies that promote digital transformation and provide competitive advantages. It describes an enterprise's approach to delivering products or services by utilizing the latest digital technologies to enhance digital innovation [33].

### 2.5. E-Learning and Privacy and Security

Numerous information systems have not been developed to be protected and secure. The safety measures that can be taken through technological means are limited and must be maintained through proper management and practices. The security measures comprise procedures, policies, software, hardware functions, processes, and organizational structures [34]. The e-learning environment includes four basic components, namely, e-learning security and information governance, design of e-learning data safety procedures and policies, implementation of countermeasures, and monitoring of the security countermeasures taken for e-learning information protection [35]. The e-learning elements include the organizational feature to ensure that security execution attains its objectives. E-learning emphasizes security management and provides flexibility to users, concurrently ensuring confidentiality, availability, and integrity of the information [36]. The e-learner user behavior is also different in terms of the use of technical applications compared to the users of other e-services. Thus, e-learning specifically needs to increase the security and privacy of information [37]. Security and privacy are significant parts of the entire secure information system. A structure whereby information security cannot be administered is not safe, regardless of the excellence of the suggested controls and their ability to provide benefits. E-learning security management is becoming complex due to the significant number of possible information privacy warnings linked to Internet use [38]. E-learning helps to provide information relating to peripheral intrusion issues by utilizing means such as XSS (or Cross-Side Scripting) and SQL injection in the site address (URL SQL injection), and performing various searches by means of search engines to obtain personalized website data similar to password cracking, session hijacking, session prediction, usernames and passwords, etc. [39]. For e-learning settings to work, it is essential that security and privacy can function. Hence, fundamental security needs, such as confidentiality, availability, and integrity, must be guaranteed when firms are learning and performing in such an environment [5].

**H1:** E-learning is positively associated with privacy and security for electronics manufacturing firms.

### 2.6. Mediating Role of the Adoption of Blockchain Technology

Recently, many firms have been adopting the latest e-learning technologies due to their numerous benefits, including transparency, record-keeping, cost optimization, and route tracking of established records. Furthermore, researchers have declared that the privacy and security of firms are being improved due to the adoption of blockchain technology [40]. Firm management is using more blockchain applications that could be optimized through blockchain expertise; accordingly, they also need to increase their understanding of blockchain technology [41]. Adoption of blockchain technology enhances product transparency and transactions by developing means that could be efficient for improved strategic planning and linkage between outsourcing, suppliers, subcontractors, and customers, thereby raising overall effectiveness [42].

Scholars have identified that blockchain technology adoption permits digitalization, automation, streamlining, and simplifying of e-learning processes, allowing management to formulate better privacy and visibility throughout their business practices. However, blockchain adoption is critical if firms wish to enhance their firm privacy and security strategy [43]. Prior researchers have demonstrated that blockchain technology adoption plays a critical role in increasing e-learning digital performance and enhancing the privacy and security policies of firms [44]. According to prior studies, the firm can make radical changes due to blockchain technologies and adoption of applications that can improve business e-learning competitive performance [45].

Furthermore, it is clear that e-learning provides the basis for blockchain adoption, which will help improve the effectiveness of firm data privacy and security. Firms have been frequently exploring ways of increasing their e-learning [4]. In blockchain adoption,

real-time knowledge and data can be exchanged among firm members, and the information comprises skills relating to the best performance and potential of privacy and security [7]. E-learning deals with the accomplishment of the latest digital skills required to enhance the competency of adoption of blockchain technology, which simultaneously accelerates the privacy and security of firms [1].

**H2:** Adoption of blockchain technology mediates between e-learning and privacy and security links.

*2.7. Digital Orientation Moderates*

Digital orientation is an important aspect of successful organization privacy and security strategies, which are also summarized in the privacy and security paradigm [15]. E-learning is a digital learning method that enables firms to obtain the most up-to-date technologies, or at least improve their understanding and knowledge [14]. Digital orientation is a primary organization-level paradigm/prototype that demonstrates a firm's capability concerning technological facts, implementation, and adoption of the latest digital technologies for the organization's success [45]. In the era of Industry 4.0, to ensure competitive benefits, an organization has to continuously revise its learning about the latest technologies. This technical learning is closely related to knowledge exploitation, integration, and exploration since technological learning increases the privacy and security of a firm [46]. In short, digital orientation has the potential to strengthen the association between e-learning and privacy and security. E-learning provides information regarding the latest technologies with conventional components and focuses on privacy and security strategies, for which digital orientation addresses the regular procedures in numerous contexts [47]. Digital orientation is regarded as the overall strategic attitude of a firm. From the perspective of firm growth, digital orientation is considered a guide for firm management to adopt the latest blockchain applications that increase the privacy and security of firm operations as the industry environment changes [48]. Digital orientation is considered to be practical, helpful, and proactive for business people to take essential steps to improve e-learning practices and design innovative products/services procedures. Through the assistance of digital orientation, the firm will be capable of benefitting from high-risk opportunities [49]. Therefore, digital orientation influences the association between e-learning and the privacy and security of SMEs. Accordingly, we hypothesize that:

**H3:** E-learning and privacy and security are moderated through digital orientation.

The Figure 1 visualizes the concept of the paper.

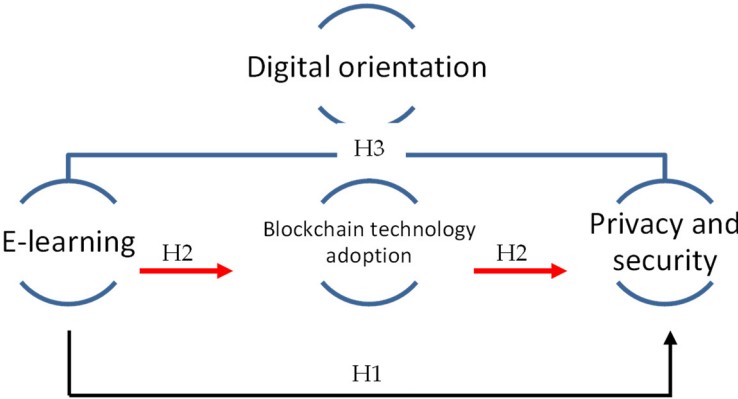

**Figure 1.** Theoretical framework. Source: Authors' concept.

**3. Materials and Methods**

The detailed methodology of the paper is presented below:

### 3.1. Research Design

The current research followed a positivist approach, which permits empirical facts from information-driven hypotheses. Additionally, comprehensive quantitative examinations were also carried out. Electronics manufacturing firms were the ideal subject matter for the current research because these firms have been greatly interested in e-learning and training initiatives that encourage their privacy and security through the adoption of blockchain technologies.

### 3.2. Data Collection

The current research followed a quantitative approach with a cross-sectional design. The random sampling technique was used to collect data from the target population of electronics manufacturing firms in Pakistan. The study sample comprises managers, owners, CEOs, and senior employees of electronics manufacturing firms. After stratifying the sample, scholars visited the electronics manufacturing firms and distributed questionnaires with the help of research associates. Twelve firms were selected based on the data collection method (details are attached in Appendix A). All the selected firms were using blockchain technology either in supply chains, payment processes, or data storage. Several questionnaires were distributed in hard copy and electronically, with a total of 450 self-administered questionnaires sent to respondents from these firms. Out of the 450 questionnaires, 315 were returned and considered valid, resulting in a return rate of 70%. Any incomplete questionnaires were discarded. Prior to distributing the questionnaires, they were tested by academia and experts who were informed about the research's scope and objectives. Their responses were considered a reliable verification of the research's mechanism. The questionnaires were divided into two sections. Section 1 included demographic information about the participants, while the next section contained items related to the study variables. Approximately 56.8% of respondents were males under the age of 35, while the other 44.2% of participants were females between the ages of 25 and 35 years.

### 3.3. Measurement

For the measurement of the study constructs, i.e., e-learning (independent variable), adoption of blockchain technology (mediator), digital orientation (moderator), and privacy and security (dependent variable), different item scales were adapted from prior studies. Initially, an email was sent to these authors requesting permission to use the scale. All authors granted us permission; however, one of the authors requested that the scale not be attached to the paper. Therefore, the scale was not included with the manuscript. To measure the validity and reliability, a pretest was performed, and five-point Likert-scales were used, that is, with a range from 1 = strongly disagree to 5 = strongly agree.

**E-learning**—To assess e-learning, a six-item scale was employed, which was adapted from [50]. This particular measure evaluates the extent to which e-learning contributes to achieving intended goals and enhancing the firm's performance. The sample question was "The e-learning user interface provides me sense of the both ease and usability".

**Adoption of blockchain-technology**—The adoption of the blockchain-technology was measured through a four-item scale which was adapted from [51]. This construct measures how different applications of blockchain increase security and privacy of the firm. An example item is "I consider our firm must put into practice blockchain technologies in upcoming future".

**Digital orientation**—To measure the digital orientation, a four-item scale was adapted from [52]. This construct measures the firm's ability to take up digital practices and use the most up-to-date technologies through proper orientation. Digital technologies assist firms to attain the advantages of the first mover in their respective industry. An example item is "new digital technology is readily accepted in our organization".

**Privacy and security**—For the measurement of privacy and security, four items were used, which were adapted from [53]. An example item is "The internet user's privacy was greatly violated".

## 4. Results

The technique developed by Fornell and Lacker (1981) [54] was utilized to assess the discriminant validity. The results of FL, Cronbach alpha ($\alpha$), and AVE are presented in Table 1, which indicate that both discriminant validity and convergent validity were achieved. Specifically, the results for CR and AVE, as displayed in Table 1, exceeded the established cutoff points. This means that the convergent reliability was above 0.70, AVE was greater than 0.50, and CR was higher than AVE.

**Table 1.** CR, factor loading and average variance extracted.

|  | Items | FL | Cronbach's Alpha | CR | AVE |
|---|---|---|---|---|---|
| E Learning | 6 | 0.72–0.75 | 0.84 | 0.94 | 0.76 |
| Blockchain Technology | 4 | 0.74–0.88 | 0.87 | 0.96 | 0.72 |
| Digital Orientation | 4 | 0.71–0.87 | 0.85 | 0.98 | 0.74 |
| Privacy and Security | 4 | 0.75–0.89 | 0.82 | 0.92 | 0.78 |

Source: Authors' computation.

As shown in Table 2, according to Joreskog and Sorbom (1996) [55], construct validity was examined through Confirmatory Factor Analysis (CFA). This research suggested a four-factor model (MO) with other substitute models where F1 = E-Learning (EL), F2 = Blockchain Technology (BCT), F3 = Digital Orientation (DO), and F4 = Privacy and Security (PS) were considered as individual factors. The outcomes shown in Table 2 proved that the four-factor model had a good fit to the data and the CFA model showed the data were accepted ($\chi^2$ = 1055.32, df = 450; $\chi^2$/df = 2.294; RMSEA = 0.08; CFI = 0.96; GFI = 0.95).

**Table 2.** Confirmatory Factor Analysis (CFA).

| Model Description | $\chi^2$ | Df | $\chi^2$/df | RMESA | GFI | CFI |
|---|---|---|---|---|---|---|
| Hypothesized-four-factor model | 1055.32 | 450 | 2.294 | 0.08 | 0.95 | 0.96 |
| Three-factor model | 1145.57 | 385 | 2.976 | 0.13 | 0.85 | 0.86 |
| Two-factor model | 1280.42 | 380 | 3.370 | 0.18 | 0.74 | 0.75 |

Source: Authors' computation.

Table 3 indicates the results of correlation. E-Learning, Blockchain Technology, Digital Orientation, and Privacy and Security. E-Learning is positively significant associated with Privacy and Security (r = 0.24 **, p = sig). Blockchain Technology is positively associated with Privacy and Security (r = 0.34 **, p = sig. DO is significantly correlated with Privacy and Security (r = 0.36 **, p = sig). The VIF scores also confirm that multi-collinearity was not a concern in this study as its significance was less than 10.0.

E-Learning positively predicts Privacy and Security. The impact of E-Learning on Privacy and Security was assessed using structural equation modeling. The findings, as shown in Table 4, confirm a significant positive relationship between E-Learning and Privacy and Security, as evidenced by the analytical proof ($\beta$ value = 0.32 **, and H1 being supported with $p \leq 0.001$).

Blockchain Technology mediates between E-Learning and Privacy and Security. The method of Preacher and Hayes (2008) [56] was utilized to inspect the mediation effect of Blockchain Technology between E-Learning and PS (EL→BCT→PS). Table 5 explores the indirect effect, and it is proved that Blockchain Technology acts as a mediator ($\beta$ = 0.1865, Lower = 0.2675 to Upper = 0.3870). Therefore, H2 was confirmed and it is established that the E-Learning and Privacy and Security relationship is mediated by Blockchain Technology.

**Table 3.** Results of Mean, SD, and Correlations.

| Variable | | Mean | SD | 1 | 2 | 3 | 4 | 5 | 6 | 7 |
|---|---|---|---|---|---|---|---|---|---|---|
| 1 | Business-Size | 3.00 | 1.02 | 1.00 | | | | | | |
| 2 | Respondent Experience | 1.34 | 0.34 | 0.013 | 1.00 | | | | | |
| 3 | Respondent Education | 1.14 | 0.35 | 0.025 | 0.028 | 1.00 | | | | |
| 4 | E Learning | 3.21 | 0.26 | 0.106 ** | 0.012 | 0.026 | 1.00 | | | |
| 5 | Blockchain Technology | 3.85 | 0.38 | −0.014 | 0.052 * | 0.034 ** | 0.017 | 1.00 | | |
| 6 | Digital Orientation | 3.32 | 0.32 | −0.022 | 0.065 * | 0.32 ** | 0.26 ** | 0.182 ** | 1.00 | |
| 7 | Privacy and Security | 1.15 | 0.24 | 0.015 | 0.001 | −0.02 | 0.247 ** | 0.348 ** | 0.365 ** | 1.00 |

Note: $* \geq 0.05$; $** >\geq 0.01$. Source: Authors' computation.

**Table 4.** Hypothesis testing of H1.

| Model | Hypothesis Description | B | F | T | Sig | Remarks |
|---|---|---|---|---|---|---|
| Model # 01 | E-Learning to Privacy and Security | 0.32 | 18.045 | 0.1135 | 0.001 | Accepted |

Source: Authors' computation.

**Table 5.** Mediating impact of Blockchain Technology between E-Learning and Privacy and Security.

| Model Detail | Data | Boot | SE | Lower | Upper | Sig |
|---|---|---|---|---|---|---|
| EL→BCT→PS | 0.1865 | 0.2460 | 0.42 | 0.2675 | 0.3870 | 0.001 |

Source: Authors' computation.

Digital Orientation moderates between E-Learning and Privacy and Security. Hierarchal regression analysis was used to explore the influence of E-Learning on Privacy and Security through the moderating role of Digital Orientation. Table 6 shows the moderating role of Digital Orientation between E-Learning and Privacy and Security. The results prove that Digital Orientation was a positive moderator between EL and PS, i.e., β = 0.36 **, $p < 0.001$. Hence, H3 was approved.

**Table 6.** Hierarchal regression outcomes for moderating influence of Digital Orientation.

| Privacy and Security | | | | | | |
|---|---|---|---|---|---|---|
| Detail | Beta | T Value | Beta | T Value | Beta | T Value |
| Step-1 | | | | | | |
| Business-age | 0.03 | 0.24 | 0.05 | 1.24 | 0.04 | 0.26 |
| Business-size | 0.02 | 0.20 | 0.04 | 0.42 | 0.16 | 0.57 |
| Respondent-education | 0.12 | 0.27 | 0.15 | 0.17 | 1.05 | 2.44 |
| Respondent-experience | 0.11 | 0.21 | 0.16 | 0.65 | 0.04 | 0.16 |
| Step-2 | | | | | | |
| E-Learning | | | 0.32 * | 5.45 | 0.32 * | 3.45 |
| Digital Orientation | | | 0.26 * | 4.52 | 0.34 * | 5.47 |
| Step-3 | | | | | | |
| EL*DO | | | | | 0.36 ** | 3.21 |
| F | | 3.75 ** | | 18.36 * | | 11.45 * |
| R2 | | 0.05 | | 0.22 | | 0.24 |
| R2 | | | | 0.26 | | 0.04 |

Notes * $p < 0.0001$, ** $p < 0.05$ (two tailed); and results of VIF were below than the threshold level. Source: Authors' computation.

## 5. Discussion

According to the aforementioned findings, the discussion of the current research is as follows. Firstly, e-learning is intrinsic to privacy and security because of their responsive and normative nature. H1 proposed that e-learning has a positive linkage with privacy and security for electronics manufacturing firms in Pakistan. The findings show that e-learning was established to be directly correlated with privacy and security. The outcomes of H1 support prior researchers who have found that numerous information systems have not

been developed to be protected and secure. The safety measures that can be taken through technological means are limited and must be maintained through proper management and practices. Security measures comprise procedures, policies, software, hardware functions, processes, and organizational structures [34].

The e-learning environment includes four basic components, namely, e-learning security and information governance, design of e-learning data safety procedures and policies, implementation of countermeasures, and monitoring of the security countermeasures taken for e-learning information protection [35]. The e-learning elements include the organizational features to ensure that security execution attains its objectives. E-learning emphasizes security management and provides flexibility to users, concurrently ensuring confidentiality, availability, and integrity of the information [36].

The e-learner user behavior and the use of technical applications also differ from those of users of other e-services. Thus, e-learning specifically needs and increases the security and privacy of information [37]. Security and privacy are significant parts of an entire secure information system. A structure whereby information security cannot be administered is not safe, regardless of the excellence of the suggested controls and their ability to provide benefits. E-learning security management is becoming complex due to the significant number of possible information privacy warnings linked to Internet use [38]. Thus, the data support H1.

H2 suggests that the adoption of blockchain technology plays a mediating role in the relationship between e-learning and privacy and security. According to research findings, it is supposed that blockchain technology adoption mediates the linkage between e-learning and privacy and security. The results are congruent with previous studies that found that many firms are using the latest e-learning technologies due to their large number of benefits. These benefits include transparency, record keeping, cost optimization, and route tracking of established records. Furthermore, researchers have declared that the privacy and security of a firm has improved due to blockchain technology adoption [40]. The firm management is using more blockchain applications that could be optimized through blockchain expertise; accordingly, firms also need to increase their understanding of blockchain technology [41]. Adoption of blockchain technology enhances product transparency and transactions by developing means that could be efficient for improved strategic planning and linkages between outsourcing, suppliers, subcontractors, and customers, thereby raising overall effectiveness [42]. Scholars have identified that blockchain technology adoption permits digitalization, automation, streamlining, and simplifying of e-learning processes that allow management to formulate better privacy and visibility throughout the electronics manufacturing business practices. However, blockchain adoption is critical if firms are willing to enhance their firm privacy and security strategy [43]. Prior researchers have demonstrated that blockchain technology adoption plays a critical role in increasing e-learning digital performance and enhancing privacy and security policies of firms [44]. According to prior studies, a firm can make radical changes due to blockchain technologies and adoption of applications that can improve business e-learning competitive performance [45]. Furthermore, it is clear that e-learning provides the basis for blockchain adoption that will increases the effectiveness of firm data privacy and security. Firms have been exploring ways by which they can increase their e-learning [4]. The findings prove H2, that is, the adoption of blockchain technology mediates between e-learning and privacy and security for electronics manufacturing firms.

Furthermore, in H3, it is supposed that digital orientation moderates the association between e-learning and privacy and security of electronics manufacturing firms. The outcomes support the prior research which finds that e-learning is a digital learning method that enables firms to obtain the most up-to-date technologies, or at least increase their understanding and knowledge [14]. Digital orientation is a primary organization-level paradigm/prototype that demonstrates the firm's capability concerning technological facts, implementation, and adoption of the latest digital technologies for the organization's success [45]. In the era of Industry 4.0, to ensure competitive benefits, an organization has to

continuously revise its learning about the latest technologies. This technical learning is closely related to knowledge exploitation, integration, and exploration, because technological learning increases the privacy and security of the firm [46]. In short, digital orientation has the potential to strengthen the association between e-learning and privacy and security for electronics manufacturing firms. E-learning provides information regarding the latest technologies with conventional components and focuses on privacy and security strategies, for which digital orientation addresses the regular procedures in numerous contexts [47]. Digital orientation is regarded as the overall strategic attitude of a firm. In the perspective of firm growth, digital orientation is considered a guide for electronics manufacturing firms' management to adopt the latest blockchain applications that increase the privacy and security of firm operations as the industry environment changes [48]. This evidence confirms H3, that is, digital orientation moderates between e-learning and privacy and security of electronics manufacturing firms.

### 5.1. Theoretical Implications

This study contributes to the knowledge of prior research and provides the following theoretical implications. Firstly, e-learning has the potential to enhance the security management of electronics manufacturing firms, e-governance, and financial markets, resulting in improved privacy and security of the firm. E-learning can have a significant influence on the privacy and security of electronics manufacturing firms. There are e-learning technologies in which operations are transparent, unalterable, and safe, and they are secured against tampering. However, e-learning is rapidly adopting blockchain technologies and becoming proficient in overcoming bottlenecks and issues.

Secondly, the use of blockchain technology is rapidly increasing and becoming popular. Consequently, the current research focuses on the role that blockchain technology may play as a mediator between e-learning and the privacy and security of electronics manufacturing firms. E-learning provides a convenient and appropriate assessment of blockchain technology in the electronics manufacturing sector, which consequently increases the privacy and security of the firm.

Thirdly, digital orientation has the potential to assist firms in introducing mindfulness programs for e-learning technology that focus on the privacy and security of the firm. Digital orientation could support electronics manufacturing firms in designing activities that emphasize e-learning due to its potential to ensure privacy and security gains. This study's findings are an absolute requirement and will be helpful for the enhancement of privacy and security in electronics manufacturing firms.

### 5.2. Practical Implications

The current research provides several practical implications for administration, policymaking, and management of electronics manufacturing firms. First, this study recommends that electronics manufacturing firms should place emphasis on enhancing their e-learning capabilities as a strategic means to increase the potential for blockchain technology adoption, which can lead to higher data security and privacy. Secondly, we suggest that blockchain technology adoption is an imperative factor in developing techniques and achieving measures that help in the privacy and security management of the firm through the antecedents of e-learning. The blockchain technology basically focuses on performance difficulties, distributed agreement practices, and system architectures. Finally, we propose that e-learning is a guiding practice for the adaptation of blockchain technological applications, and correspondingly, digital orientation helps in acquiring means that support the privacy and security management of electronics manufacturing firms.

### 5.3. Limitations and Directions

Several limitations exist in this study, which could be explored as potential directions for future research. Like several other studies, the current research is not without limitations. The current study focuses on electronics manufacturing firms, and a quantitative method

was used to evaluate the effectiveness of privacy and security in the firm. Cross-sectional data were employed to achieve the research purposes. It was challenging to gather data from owners and management of electronics manufacturing firms; therefore, the study sample size is small. Only one site was used for data collection from participants in this study. Accordingly, we recommend that future research should be carried out in a variety of other national contexts, including economic regions outside of the electronics manufacturing industry. Moreover, these economic segments should offer exciting opportunities for future studies, which may encompass privacy and security in social ventures. The enormous potential of blockchain technology and its problem-solving competency will drastically change the setting of efforts to acquire long-term security management applications. We propose that management and policymakers should integrate other constructs and investigate their impacts on privacy and security management in the firm.

**Author Contributions:** Conceptualization, A.A.N. and A.G.; methodology, M.H. and R.A.T.; software, A.J.; validation, A.G.; and A.A.N.; formal analysis, A.A.N.; investigation, M.H., resources, R.A.T. and Z.Y.; data curation, M.H.; writing—original draft preparation, A.J. and R.A.T.; writing—review and editing, A.G. and Z.Y.; visualization, A.A.N.; supervision, A.G. and A.J.; project administration, Z.Y.; funding acquisition, R.A.T. All authors have read and agreed to the published version of the manuscript.

**Funding:** Researchers Supporting Project number (RSP2023R87), King Saud University, Riyadh, Saudi Arabia.

**Institutional Review Board Statement:** Study was conducted following the guidelines of Declaration-of-Helsinki. It is approved by Ethics Committee of GCMS; DNo. 345-987.

**Informed Consent Statement:** Informed consent was obtained from participants involved in this research.

**Data Availability Statement:** Data will be provided on request.

**Conflicts of Interest:** The authors declare no conflict of interest.

## Appendix A

List of companies

1. Haier Pakistan
2. Dawlance
3. Waves Singer Pakistan
4. Orient Electronics
5. PEL
6. Gree Electric
7. Changhong Ruba
8. EcoStar
9. Sogo
10. Super Asia Group
11. Ruba Digital
12. Pak Elektron Limited (PEL)

**Appendix B**

Detail of respondents

| 1. | S. No | 2. | Designation | 3. | Number of respondents | 4. | Percentage |
|---|---|---|---|---|---|---|---|
| | | 5. | CEO | 6. | 5 | 7. | 1.58730159 |
| | | 8. | Chief Operating Officer | 9. | 4 | 10. | 1.26984127 |
| | | 11. | Chief Financial Officer | 12. | 3 | 13. | 0.95238095 |
| | | 14. | Vice President | 15. | 17 | 16. | 5.3968254 |
| | | 17. | Director | 18. | 28 | 19. | 8.88888889 |
| | | 20. | Manager | 21. | 258 | 22. | 81.9047619 |
| | | 23. | **Total** | | | 24. | **100** |

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
