# Peer review of "Leading Role of E-Learning and Blockchain towards Privacy and Security Management: A Study of Electronics Manufacturing Firms"

_electronics, doi:10.3390/electronics12071579_

Round 1

Reviewer 1 Report

You have written a quality paper but you need to address some major errors to improve the standard of the paper. I explain my concerns in more detail below. I ask that the authors specifically address each of my comments in their responses.

In the Abstract, it was written that the random sampling technique was used however, in the “Data Collection” section it was written that convenient sampling was used. It should be corrected. 

More details should be given about the participants. For example; How many participants from each electronic device firm should be given? 

How many participants are managers, owners, CEO, and senior employees of electronics manufacturing firms should be given. 

“Business age, business size, respondent education” should be given. 

The written of the “Blockchain” should be standard/same everywhere in the paper. However, in some places, it was used like “Block Chain” and in some places, it was used like “Blockchain”.

Reviewer 2 Report

Well structured work. There are some questions to the resulting part, but it does not affect quality of the work in general.

Reviewer 3 Report

I have some questions and comments about the paper.

How are electronics manufacturing companies in Pakistan using blockchain technology? Give examples.

In many places, the paper talks about the role of e-learning. Please clarify what e-learning is exactly about. Do the authors have in mind the use of a learning management system (LMS) or e-learning for the use of blockchain technologies, privacy, and security?

How do electronics manufacturing firms differ from other firms in their use of blockchain technology? Does what was said about electronics manufacturing firms also apply to other firms that have a different scope of activity, e.g. manufacture of textiles or drug production?

I believe that it would be good to publish the questionnaire that collected information about the conducted research.

I recommend outlining what are the advantages and disadvantages of using blockchain technologies in electronics manufacturing firms.

The paper should be proofread.

Round 2

Reviewer 3 Report

The authors have made efforts to improve the manuscript. I recommend publishing the work.

Author Response

Dear Editor

Manuscript ID: Electronics-2286577

The paper titled “Leading Role of E-learning, Blockchain Towards Privacy and Security Management: A Study on Electronics Manufacturing Firms” has been reviewed as per comments of the referee(s).These changes are incorporated and highlighted in the text; the details of the changes are as follows:

Reviewer 3 Comments

Reviewer 3 Changes

1. The authors have made efforts to improve the manuscript. I recommend publishing the work.

1.      Thanks for appreciation and guideline for improvement.
